# Platelet Counts and Risk of Severe Retinopathy of Prematurity: A Bayesian Model-Averaged Meta-Analysis

**DOI:** 10.3390/children10121903

**Published:** 2023-12-08

**Authors:** Mohamad F. Almutairi, Silvia Gulden, Tamara M. Hundscheid, František Bartoš, Giacomo Cavallaro, Eduardo Villamor

**Affiliations:** 1Division of Neonatology, Department of Pediatrics, MosaKids Children’s Hospital, Maastricht University Medical Center (MUMC+), School for Oncology and Reproduction (GROW), Maastricht University, 6202 AZ Maastricht, The Netherlands; 2Neonatal Intensive Care Unit, Sant’Anna Hospital, 22020 Como, Italy; 3Department of Psychology, University of Amsterdam, 1001 NK Amsterdam, The Netherlands; 4Neonatal Intensive Care Unit, Fondazione IRCCS Ca’ Granda Ospedale Maggiore Policlinico, 20122 Milan, Italy

**Keywords:** thrombocytopenia, platelets, retinopathy of prematurity

## Abstract

Background: We aimed to conduct a systematic review and Bayesian model-averaged meta-analysis (BMA) on the association between platelet counts and severe retinopathy of prematurity (ROP). Methods: We searched for studies reporting on platelet counts (continuous variable) or thrombocytopenia (categorical variable) and severe ROP or aggressive posterior ROP (APROP). The timing of platelet counts was divided into Phase 1 (<2 weeks) and Phase 2 (around ROP treatment). BMA was used to calculate Bayes factors (BFs). The BF_10_ is the ratio of the probability of the data under the alternative hypothesis (H_1_) over the probability of the data under the null hypothesis (H_0_). Results: We included 21 studies. BMA showed an association between low platelet counts and severe ROP. The evidence was strong (BF_10_ = 13.5, 7 studies) for phase 1 and very strong (BF_10_ = 51.0, 9 studies) for phase 2. Thrombocytopenia (<100 × 10^9^/L) in phase 2 was associated with severe ROP (BF_10_ = 28.2, 4 studies). Following adjustment for publication bias, only the association of severe ROP with thrombocytopenia remained with moderate evidence in favor of H_1_ (BF_10_ = 4.30). Conclusions: Thrombocytopenia is associated with severe ROP. However, the evidence for this association was tempered when results were adjusted for publication bias.

## 1. Introduction

Retinopathy of prematurity (ROP) is a serious complication of preterm birth that can cause severe vision loss or blindness if untreated [1,2]. Despite improvements in neonatal care, ROP remains challenging for neonatologists and ophthalmologists because it requires early detection and intervention to prevent visual impairment. In order to develop effective screening strategies and treatment plans, a comprehensive understanding of the risk factors and underlying mechanisms of ROP is essential [1,2].

The etiopathogenesis of ROP is multifactorial and involves both intrinsic and environmental factors. The two main risk factors for developing ROP are prematurity degree and supplemental oxygen exposure [1,2]. In addition, many other factors, such as biological sex, genetic predisposition, perinatal infection/inflammation, or pre- and postnatal malnutrition, have been associated with the risk of developing ROP [1,2,3,4,5].

Data from both pre-clinical and clinical studies suggest that platelets may play a role in the pathogenesis of ROP [6,7,8,9,10]. The emerging role of platelets as carriers of key angiogenic regulatory proteins in their α-granules is the main rationale for platelet involvement in the development of ROP [8]. A growing number of cohort and case-control studies have examined the potential association between platelet counts and the risk of ROP. Data from some of these studies suggest an association between thrombocytopenia and the risk of developing severe ROP [6,7,9,10]. Seliniotaki et al. conducted a systematic review in 2022 that included 19 studies on this topic [8]. Although they found evidence for thrombocytopenia as a risk factor for ROP, the heterogeneity in study design, subject characteristics, case and control definitions, time points for platelet evaluation, and thrombocytopenia definition led them to decide not to carry out a meta-analysis [8]. However, some patterns of homogeneity can be observed in the studies included in the systematic review by Seliniotaki et al., as well as in studies published subsequently. This would allow a quantitative analysis of the association between platelet counts and ROP. Our current objective is to perform such a meta-analysis using a Bayesian approach.

Meta-analysis is usually carried out in the classical or frequentist framework, but Bayesian meta-analysis provides a number of advantages and has recently gained an increasing amount of interest in the biomedical sciences [11,12]. Bayesian meta-analysis is particularly appropriate when there is a small number of studies. Furthermore, Bayesian analysis allows the quantification of the evidence for two or more hypotheses. The Bayes factor (BF) is a way of quantifying the relative degree of support for a hypothesis in a data set and is the primary tool used in Bayesian inference for hypothesis testing [12,13,14,15,16]. Specifically, one may obtain evidence in favor of the null hypothesis (H_0_), evidence in favor of the alternative hypothesis (H_1_), or absence of evidence (when both hypotheses predict the data about equally well) [12,13,14,15,16]. In contrast, the *p*-value from classical frequentist methods cannot discriminate evidence of absence from absence of evidence [12,13,14,15,16]. Therefore, the Bayesian framework can provide a broader and arguably more informative set of interpretations compared with classical frequentist analysis.

## 2. Materials and Methods

The methodology of this study is based on previous meta-analyses conducted by our group on thrombocytopenia as a risk factor for patent ductus arteriosus [17] and risk factors for ROP [3,4,5]. The study was conducted and reported according to the PRISMA (Preferred Reporting Items for Systematic Reviews and Meta-Analyses) and MOOSE (Meta-Analyses of Observational Studies in Epidemiology) guidelines. The protocol of the review was registered in the PROSPERO International Register of Systematic Reviews (ID = CRD42021248183). The research question was “Are lower platelet counts associated with an increased risk of developing severe ROP in very and extremely preterm infants?”

### 2.1. Sources and Search Strategy

We searched PubMed, Embase, and Web of Science databases. The details of the search strategy are depicted in Appendix A. The literature search was updated up to June 2023.

### 2.2. Study Selection and Definitions

We included studies if they had a prospective or retrospective cohort or case-control design, examined very preterm (GA ≤ 32 weeks) or very low birth weight (<1500 g) infants, and included data on the association between platelet counts or other platelet parameters and rate of severe ROP. Severe ROP was defined as prethreshold disease type 1 according to the ETROP criteria, or as any ROP requiring treatment [18]. Aggressive posterior ROP (APROP) was analyzed as a separate category [19].

With regard to the timing of platelet determinations, and because ROP is a two-phase disease, two time periods were established to categorize the information. Phase 1 included platelet counts performed in the first 2 weeks of life or before 30 weeks postmenstrual age (PMA). Phase 2 included platelet counts performed after 4 weeks of age, 30 weeks PMA, or in the days surrounding the diagnosis and/or treatment of ROP.

### 2.3. Extraction od Data and Study Quality Assessment

Three investigators (MA, TH, EV) extracted data on characteristics of the studies, platelet counts, and rates of ROP and thrombocytopenia. A second group of investigators (SG, GC) checked the data extraction for completeness and accuracy. The methodological quality of the included studies was assessed using the Newcastle–Ottawa Scale (NOS) for cohort or case-control studies [20].

### 2.4. Statistical Analysis

The effect size of dichotomous variables (e.g., thrombocytopenia) was expressed as log odds ratio (logOR), while the effect size of continuous variables (e.g., platelet counts) was expressed using Hedges’ *g*. The values of logOR or Hedges’ *g* and the corresponding standard errors of each individual study were calculated using comprehensive meta-analysis V4.0 software (Biostat Inc., Englewood, NJ, USA). The results were further pooled and analyzed using a Bayesian model-averaged (BMA) meta-analysis [15,16]. The BMA was performed in JASP, which utilizes the metaBMA R package [21,22]. BMA employs BFs and Bayesian model-averaging to evaluate the likelihood of the data under the combination of models assuming the presence vs. the absence of the meta-analytic effect and heterogeneity [15,16]. The BF_10_ is the ratio of the probability of the data under H_1_ over the probability of the data under H_0_. For the interpretation of the BF_10_, we used the evidence categorization described by Lee & Wagenmakers [23] (Figure 1). The BF_rf_ is the ratio of the probability of the data under the random effects model over the probability of the data under the fixed effects model. The categorization of the evidence in favor of the random effects (BF_rf_ > 1) or the fixed effects (BF_rf_ < 1) was similar to the one used for BF_10_. We used the empirical prior distributions based on neonatal studies from the Cochrane Database of Systematic Reviews [16,19,20]; i.e., prior distributions for continuous outcomes (Hedges’ *g*) corresponded to mu ~ Student’s *t* (µ = 0, σ = 0.42, ν = 3), tau ~ Inverse-Gamma (k = 1.68, θ = 0.38), while prior distributions for dichotomous outcomes (logOR) corresponded to mu ~ Student’s *t* (µ = 0, σ = 0.29, ν = 3), tau ~ Inverse-Gamma (k = 1.80, θ = 0.42) [12,15,16].

We used robust Bayesian meta-analysis (RoBMA) to assess the robustness of the results to the potential presence of publication bias [24]. RoBMA extends the Bayesian model-averaged meta-analysis by the two major publication bias adjustment techniques: selection models (adjusting for the publication bias operating on *p*-values) [25] and precision-effect test and precision-effect estimate with standard errors (PET-PEESE, adjusting for the relationship between effect sizes and standard errors) [26]. The resulting RoBMA ensemble contains 36 models composed of the following assumptions about the presence vs. absence of the effect (2×), presence vs. absence of between-study heterogeneity (2×), and presence vs. absence of publication bias adjustment models (6 selection models, PET, PEESE, and no bias). We used RoBMA with the same prior distributions for the effect and heterogeneity as in BMA and the default prior distributions for the publication bias adjustment part. Publication bias was expressed as BF_bias_ using the same categories for evidence previously described for BF_10_ and BF_rf_.

## 3. Results

### 3.1. Characteristics of the Studies and Risk of Bias Assessment

The search process PRISMA flow diagram is depicted in Appendix A. Of 2645 studies with potential relevance, 21 were included [6,7,9,10,27,28,29,30,31,32,33,34,35,36,37,38,39,40,41,42,43]. These studies reported on 3625 infants. The characteristics of the included studies are shown in Appendix A. The risk of bias assessment is depicted in Appendix A. Twenty studies received scores above seven points, indicating a low risk of bias. One study [39] was only published as an abstract, and risk of bias could not be fully assessed.

### 3.2. Bayesian Meta-Analysis

Table 1 summarizes the results of the BMA. Regarding platelet counts as the continuous variable, BMA showed that the evidence in favor of H_1_ (presence of association between a lower platelet count and severe ROP) was strong in phase 1 (BF_10_ = 13.5, 7 studies, Figure 2A) and very strong in phase 2 (BF_10_ = 51.0, 9 studies, Figure 2B). Regarding thrombocytopenia (<100 × 10^9^/L), BMA showed moderate evidence in favor of H_1_ (association with severe ROP) in phase 1 (BF_10_ = 6.01, 3 studies, Figure 3A) and strong evidence in phase 2 (BF_10_ = 28.2, 4 studies, Figure 3B). In addition, the BMA showed strong evidence in favor of an association between severe ROP and platelet transfusions (BF_10_ = 12.0, 5 studies, Figure 3C). The BMA showed inconclusive evidence in favor of H_0_ for the association between severe ROP and both mean platelet volume (MPV) and Platelet Mass Index (PMI) in both phases (Figure 4). Regarding APROP, the BMA showed inconclusive evidence for the association with platelet counts, thrombocytopenia, and platelet transfusions (Table 1, Figure 5).

The RoBMA results are shown in Table 2. The RoBMA showed moderate evidence in favor of publication bias for the associations between severe ROP and platelet counts in phase 1 (BF_bias_ = 3.28) and between APROP and thrombocytopenia (BF_bias_ = 8.98). In addition, RoBMA showed strong evidence in favor of publication bias for the associations of severe ROP with platelet counts in phase 2 (BF_bias_ = 12.40) and platelet transfusions (BF_bias_ = 11.65). Following adjustment for publication bias, the RoBMA showed moderate evidence in favor of H_1_ for the association between severe ROP and thrombocytopenia (BF_10_ = 4.30, 4 studies, Table 2). In all other meta-analyses, the adjustment for publication bias tempered the evidence to inconclusive (Table 2).

## 4. Discussion

This is the first meta-analysis of the association between platelets and ROP. The BMA showed moderate to strong evidence in favor of an association between low platelet counts and the risk of developing severe ROP. However, association does not imply causation. Low platelet counts may be merely a proxy for some pathological conditions, such as infectious-inflammatory events, which have been demonstrated to drive the increased risk of developing severe ROP. In addition, the RoBMA showed that there was moderate to strong evidence of small study effects/publication bias in many of the meta-analyses. The evidence for an association between low platelet counts and ROP was tempered when the results were adjusted for such bias, but the RoBMA found moderate evidence in favor of an association of late thrombocytopenia with the risk of severe ROP.

The small study effect is a generic term referring to the phenomenon that smaller trials may show different, and often larger, effects than larger trials [24,44,45,46]. This inverse relationship between study size and effect size may be an indication of non-reporting or publication bias. This is because studies with “statistically significant” results are often more likely to be submitted by authors and published by journals than studies with “non-significant” results [24,44,45,46]. Unfortunately, because we cannot know the unreported results or the exact mechanism of omission, a perfect solution to the problem of potential missing studies is impossible [24]. Funnel plots have long been used to assess the possibility of missing results in a meta-analysis [46]. However, statistical tests based on funnel plots require careful interpretation, especially when the number of studies included in the meta-analysis is small. In fact, in the Cochrane Handbook for Systematic Reviews of Interventions, the authors state that “as a rule of thumb, tests for funnel plot asymmetry should be used only when there are at least 10 studies included in the meta-analysis, because when there are fewer studies the power of the tests is low” [46]. None of the meta-analyses included in our study would be suitable for publication bias analysis according to this recommendation.

To adjust for likely publication bias from patterns observed in the reported research record, several methods are available [24]. An alternative method is to explicitly integrate the various approaches and to let the data determine the contribution of each model on the basis of its relative accuracy in predicting the observed data [24,47]. The RoBMA simultaneously applies a series of meta-analytic models to the data and estimates the effect size by taking all models into account. Therefore, the RoBMA can quantify evidence for the presence as well as the absence of publication bias and can correct for publication bias in cases where the true effect size differs between studies [24,47].

The present results of the RoBMA suggest that the association between platelets and ROP is overestimated by the included studies. The first call for attention to a possible association between platelet count and ROP came from the study by Vinekar et al. [9]. Since then, several groups have attempted to reproduce these results. It is plausible to speculate that those who did not find “statistically significant” results were discouraged from submitting them or had more difficulty getting them published. However, it should be noted that after adjusting for publication bias, there was still moderate evidence of an association between thrombocytopenia and ROP.

As mentioned in the introduction, the biological plausibility for a role of platelets in the pathogenesis of ROP is based on the capacity of activated platelets to release pro- and anti-angiogenic mediators [8]. Among these molecules are vascular endothelium growth factor (VEGF) and insulin-like growth factor-1 (IGF-1). However, one of the major difficulties in understanding the potential role of platelet-released angiogenic mediators in the pathogenesis of ROP is the biphasic nature of the disease. Phase 1 ROP is due to the cessation of vascularization and loss of normal vessels, which begins immediately after birth and is secondary to an oxygen-induced decrease in VEGF and IGF-1 [1]. Phase 2 extends from 30 to 32 weeks postmenstrual age to term. During this phase, VEGF levels increase, especially when there is retinal hypoxia with an increase in retinal metabolism and oxygen demand, leading to abnormal vascular proliferation [1]. Therefore, it is difficult to explain how, if activated platelets are VEGF releasers, a low number could favor the development of ROP in phase 2, which is characterized by a pathological increase of this mediator. In phase 1, when there are low levels of pro-angiogenic factors, a pathogenic role of thrombocytopenia in ROP would be more understandable. However, our meta-analysis suggests that the evidence in favor of the association of severe ROP with thrombocytopenia is stronger in phase 2.

Unfortunately, the absence of data has prevented us from conducting a meta-analysis on the association of severe thrombocytopenia and ROP. Thrombocytopenia in the adult is classified as mild (<150 to 100 × 10^9^/L), moderate (<100 to 50 × 10^9^/L), or severe (<50 × 10^9^/L), but the validity and clinical relevance of these values for the very and extremely preterm infant remains unclear. Mild and moderate thrombocytopenia is very common in preterm infants. It can be divided into early thrombocytopenia, which occurs within the first 72 h of life, and late thrombocytopenia, which occurs after the first 72 h of life [48,49]. Early thrombocytopenia is associated with intrauterine growth restriction (IUGR), whereas late thrombocytopenia is mainly related to sepsis and necrotizing enterocolitis (NEC) [48,49]. It is interesting to note that IUGR [50,51,52], neonatal infections [37,53,54], and NEC [53,55] are associated with an increased risk of severe ROP. This again raises the question of whether thrombocytopenia is an epiphenomenon rather than a true pathogenic factor in developing ROP. Unfortunately, the low number of studies did not allow us to conduct a subgroup analysis, a meta-regression, or any other type of sensitivity analysis that could help to answer this question. In addition, adequate assessment of potential confounders such as gestational age, birth weight, NEC, or sepsis would require complete data from each individual study to conduct a meta-analysis of individual patient data, which is beyond our scope.

Although information on the criteria for platelet transfusion was not available in the included studies, it is common clinical practice to transfuse platelets only in the most severe forms of thrombocytopenia [56]. Therefore, it could be speculated that platelet transfusion is a surrogate for severe thrombocytopenia. The BMA showed strong evidence of an association between platelet transfusion and severe ROP. However, the RoBMA found evidence of publication bias, and after adjusting for this bias, the evidence in favor of the association was downgraded to inconclusive. Of note, the clinical benefit of platelet transfusions in preterm infants is under question [57,58]. A recent randomized controlled trial compared two thresholds (50 × 10^9^/L vs. 25 × 10^9^/L) for platelet transfusion and found that infants assigned to receive platelet transfusions at the higher threshold had a higher rate of mortality or major bleeding than those who received platelet transfusions at the lower platelet count threshold [57,58]. The rate of severe ROP did not differ between the two thresholds [57,58].

Several investigators have suggested that other quantitative or qualitative platelet parameters, such as MPV or PMI, rather than platelet counts, are associated with increased risk of ROP and other prematurity complications, including sepsis, intraventricular hemorrhage, patent ductus arteriosus, or bronchopulmonary dysplasia [17,59,60,61,62]. Increased size of platelets (i.e., high MPV) is recognized as a marker of platelet activation because small platelets are less reactive than large platelets [63,64]. The PMI is the MPV multiplied by the platelet count. It therefore takes into account both platelet size and number. Nevertheless, no evidence of a possible association between severe ROP and both MPV and PMI was found in our meta-analysis.

## 5. Conclusions

In conclusion, the present data suggest a possible association between low platelet counts and severe ROP. However, our results are limited by publication bias and the fact that thrombocytopenia may be or is merely a surrogate for other conditions, such as IUGR, neonatal sepsis, or NEC, which increase the risk of developing ROP.

## Figures and Tables

**Figure 1 children-10-01903-f001:**
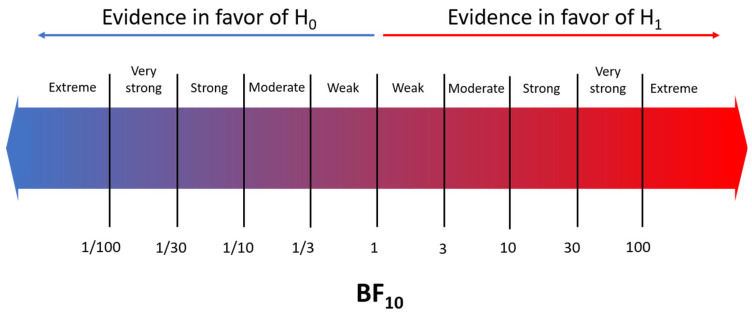
Interpretation of the Bayes factor (BF) [23].

**Figure 2 children-10-01903-f002:**
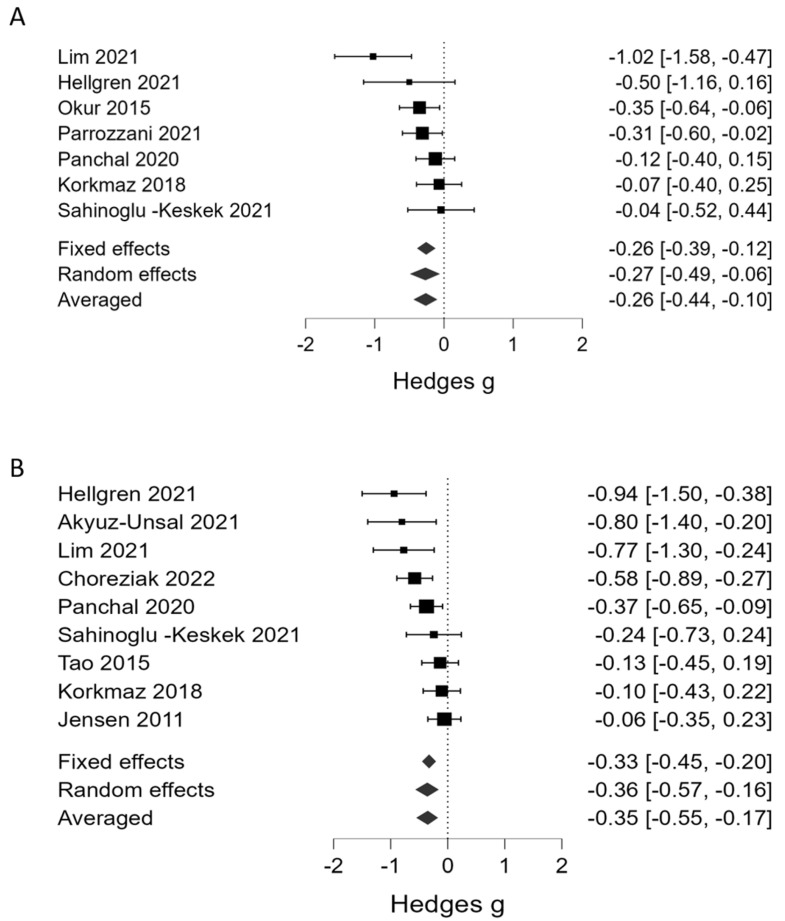
Bayesian model-averaged meta-analysis (BMA) of the association of severe ROP with platelet counts in phase 1 (**A**) and phase 2 (**B**). Hedges’ *g* < 0 indicates lower platelet counts in the ROP group [6,7,27,28,29,30,35,36,39,40,41].

**Figure 3 children-10-01903-f003:**
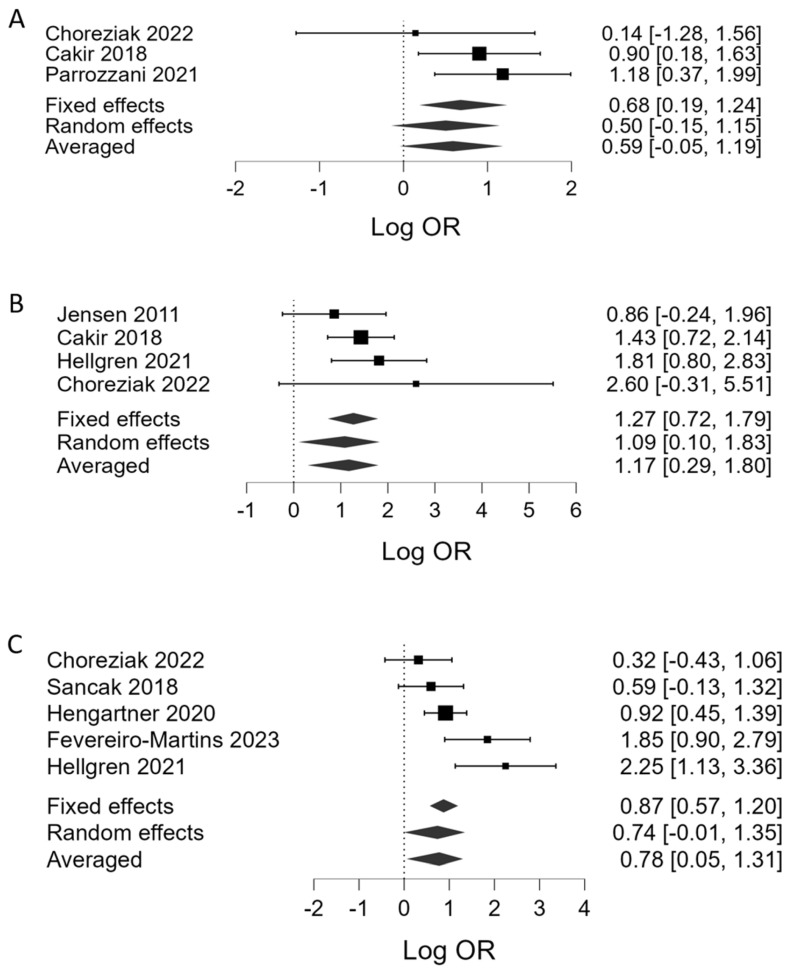
Bayesian model-averaged meta-analysis (BMA) of the association of severe ROP with (**A**) thrombocytopenia during phase 1, (**B**) thrombocytopenia during phase 2, and (**C**) platelet transfusions. Log OR > 0 indicates higher risk in the ROP group [6,7,10,27,28,32,33,38].

**Figure 4 children-10-01903-f004:**
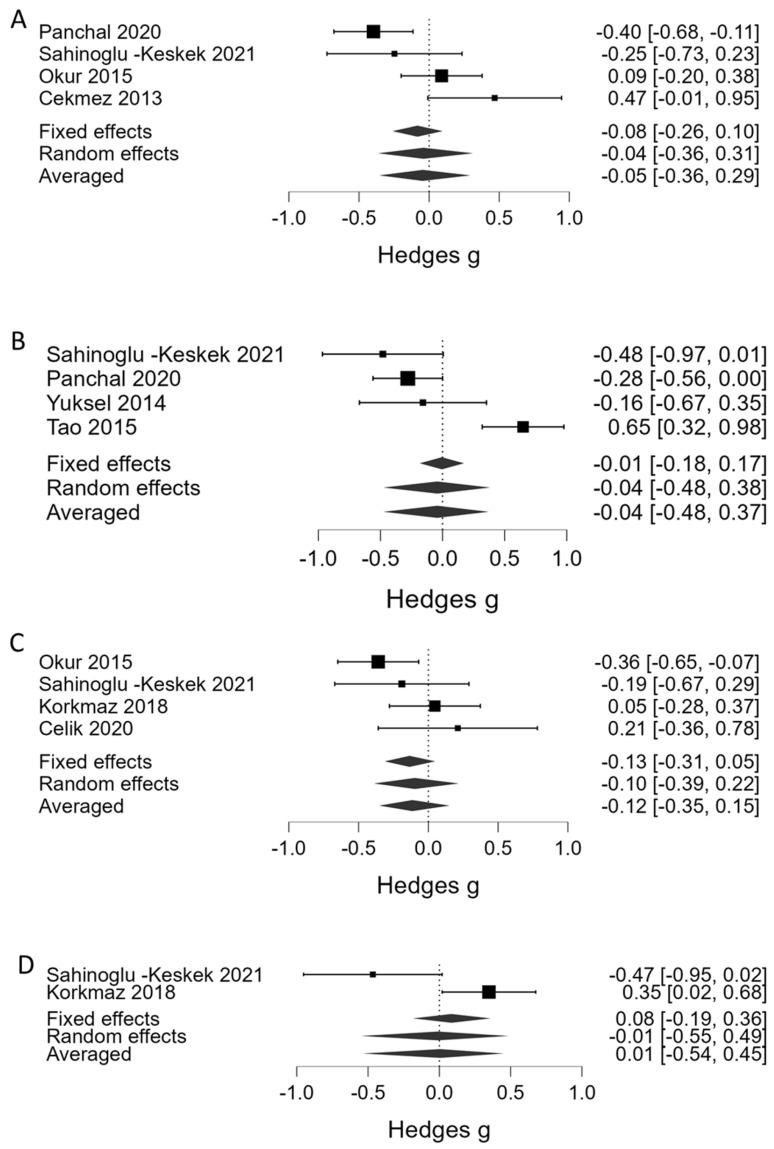
Bayesian model-averaged meta-analysis (BMA) of the association of severe ROP with mean platelet volume (MPV) in phase 1 (**A**) and phase 2 (**B**), and with Platelet Mass Index (PMI) in phase 1 (**C**) and phase 2 (**D**). Hedges’ *g* < 0 indicates lower value in the ROP group [29,31,35,39,41,42,43].

**Figure 5 children-10-01903-f005:**
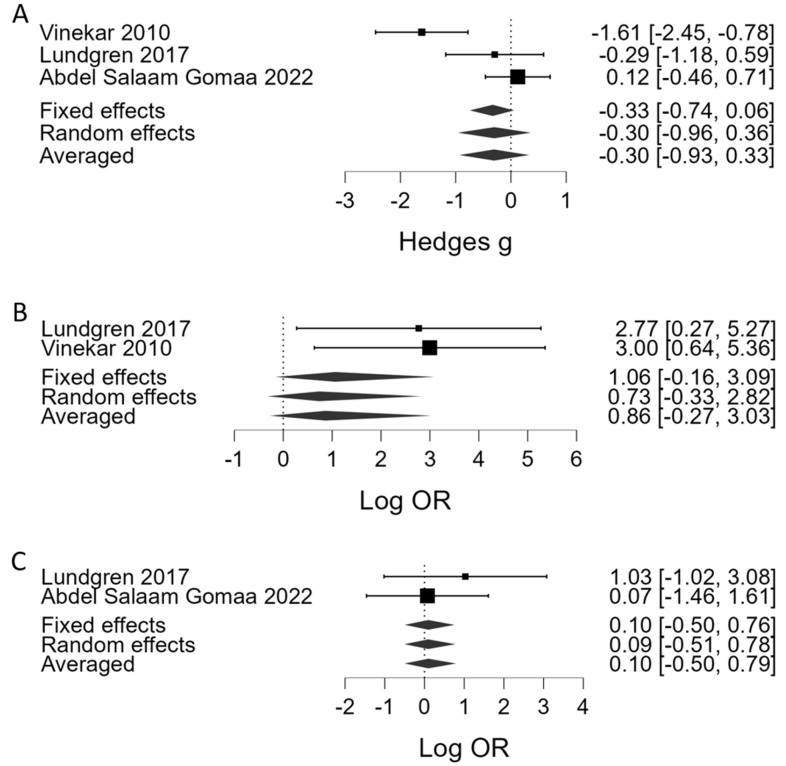
Bayesian model-averaged meta-analysis (BMA) of the association of aggressive posterior ROP (APROP) with platelet counts in phase 2 (**A**), thrombocytopenia in phase 2 (**B**), and platelet transfusions (**C**). Hedges’ *g* < 0 indicates lower value in the APROP group. Log OR > 0 indicates higher risk in the APROP group [9,34,37].

**Table 1 children-10-01903-t001:** Bayesian model-averaged meta-analysis (BMA) of the association between platelet counts and ROP.

Condition	Variable	Phase	k	Effect Size		SD	95% Credible Interval		BF_10_	Evidence for		*p*-Value Frequentist nalysis ^a^	BF_rf_	Evidence for	
							Lower Limit	Upper Limit		H_1_	H_0_			Random Effects	Fixed Effects
Severe ROP	Platelet counts	1	7	Hedges’ *g*	−0.26	0.09	−0.42	−0.10	13.5	strong		<0.001	0.79		weak
		2	9	Hedges’ *g*	−0.34	0.10	−0.55	−0.15	51.0	very strong		<0.001	2.75	weak	
	Thrombocytopenia (<100 × 10^9^/L)	1	3	Log OR	0.59	0.31	−0.03	1.19	6.01	mod.		<0.001	1.33	weak	
		2	4	Log OR	1.17	0.36	0.29	1.79	28.2	strong		<0.001	1.00	weak	
	PMI	1	4	Hedges’ *g*	−0.12	0.12	−0.35	0.15	0.49		weak	0.14	1.10	weak	
		2	2	Hedges’ *g*	0.01	0.24	−0.54	0.45	0.48		weak	0.50	3.47	mod.	
	MPV	1	4	Hedges’ *g*	−0.05	0.16	−0.36	0.29	0.36		weak	0.32	4.95	mod.	
		2	4	Hedges’ *g*	−0.04	0.21	−0.48	0.37	0.42		weak	0.88	257.0	extr.	
	Platelet transfusion	both	5	Log OR	0.78	0.30	0.06	1.30	12.0	strong		<0.001	2.72	weak	
APROP	Platelet counts	2	3	Hedges’ *g*	−0.30	0.31	−0.93	0.33	1.20	weak		0.55	6.54	mod.	
	Thrombocytopenia (<100 × 10^9^/L)	2	2	Log OR	0.86	0.88	−0.27	3.03	2.34	weak		<0.001	1.32	weak	
	Platelet transfusion	both	2	Log OR	0.10	0.32	−0.50	079	0.90		weak	0.51	0.74		weak

^a^ Random effects frequentist meta-analysis.

**Table 2 children-10-01903-t002:** Robust Bayesian meta-analysis (RoBMA) of the association between platelet counts and ROP.

Condition	Variable	Phase	k	Effect Size		95% Credible Interval		BF_10_	BF_rf_	BF_bias_
						Lower Limit	Upper Limit			
Severe ROP	Platelet counts	1	7	Hedges’ *g*	−0.18	−0.41	0.18	1.17	0.74	3.27
		2	9	Hedges *g*	−0.14	−0.46	0.32	0.64	0.95	12.4
	Thrombocytopenia (<100 × 10^9^/L)	1	3	Log OR	0.45	−0.20	1.10	2.75	1.43	1.86
		2	4	Log OR	0.83	−0.14	1.67	4.30	1.24	2.00
	PMI	1	4	Hedges’ *g*	−0.09	−0.34	0.20	0.43	0.95	0.77
		2	2	Hedges’ *g*	0.10	−0.42	0.71	0.52	2.23	1.34
	MPV	1	4	Hedges’ *g*	−0.02	−0.33	0.35	0.34	3.71	0.70
		2	4	Hedges’ *g*	0.07	−0.44	0.63	0.55	118.1	1.22
	Platelet transfusion	both	5	Log OR	0.28	−0.38	1.07	1.21	1.99	11.65
APROP	Platelet counts	2	3	Hedges’ *g*	−0.21	−0.81	0.49	1.01	4.91	1.18
	Thrombocytopenia (<100 × 10^9^/L)	3	2	Log OR	0.30	−0.54	1.86	1.20	1.10	8.98
	Platelet transfusion	both	2	Log OR	0.04	−0.63	0.76	0.90	0.85	0.70

## Data Availability

The data presented in this study are available in article and Appendix A.

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
