# Peer review of "Platelet Counts and Risk of Severe Retinopathy of Prematurity: A Bayesian Model-Averaged Meta-Analysis"

_children, 2023, doi:10.3390/children10121903_

Round 1

Reviewer 1 Report

Comments and Suggestions for Authors

The authors use Bayes statistics to analyze the correlation between ROP and platelet count. This is interesting because there are very few papers that use Bayes statistics for this type of statistical analysis. However, it is unclear why the authors used Bayes statistics in this report. I think that the following points need to be added in more detail.

1. why the authors analyze the correlation with platelet

2. what is the purpose of using Bayesian statistics

3. how confounder were eliminated

Author Response

Thank you very much for your comments and suggestions.

The rationale is that in several individual studies an association between ROP and thrombocytopenia has been reported. This idea is now more clearly reflected in the new version of the manuscript.

“Data from both pre-clinical and clinical studies suggest that platelets may play a role in the pathogenesis of ROP (5-9). The emerging role of platelets as carriers of key angiogenic regulatory proteins in their α-granules is the main rationale for platelet involvement in the development of ROP (7). A growing number of cohort and case-control studies have examined the potential association between platelet counts and the risk of ROP. Data from some of these studies suggest an association between thrombocytopenia and the risk of developing severe ROP (5, 6, 8, 9). Seliniotaki et al. conducted a systematic review in 2022 that included 19 studies on this topic (7). Although they found evidence for thrombocytopenia as a risk factor for ROP, the heterogeneity in study design, characteristics of the studied population, definition of cases and controls, time points of platelet assessment, and definition of thrombocytopenia led them to decide not to carry out a meta-analysis (7). However, some patterns of homogeneity can be observed in the studies included in the systematic review by Seliniotaki et al. (7), as well as in the studies published subsequently. This would allow a quantitative analysis of the association between platelet counts and ROP. Our current objective is to perform such a meta-analysis using a Bayesian approach.

  1. what is the purpose of using Bayesian statistics

When the number of studies is low, Bayesian statistics allows a more accurate assessment of the strength of the evidence. It also allows differentiation between absence of evidence and evidence of absence. This information is contained in the manuscript and has now been briefly expanded:

“Meta-analysis is usually performed in the classical or frequentist framework, but Bayesian meta-analysis offers several advantages and has recently gained increasing interest in the biomedical sciences (10, 11). Bayesian meta-analysis is particularly appropriate when there are a small number of studies. Furthermore, the Bayesian analysis allows evidence to be quantified for two or more hypotheses. The Bayes factor (BF) is the way to quantify the relative degree of support for a hypothesis in a data set and is the primary tool used in Bayesian inference for hypothesis testing (11-15). Specifically, one may obtain evidence in favor of the null hypothesis (H0), evidence in favor of the alternative hypothesis (H1), and absence of evidence (when both hypotheses predict the data about equally well) (11-15). In contrast, the p-value from classical frequentist methods cannot discriminate evidence of absence from absence of evidence (11-15). Therefore, the Bayesian framework may provide a wider, and arguably more informative, set of interpretations than that typically provided by a frequentist analysis.”

.

  1. how confounder were eliminated

Unfortunately, the low number of studies did not allow us to conduct a subgroup analysis, a meta-regression or any other type of sensitivity analysis. In addition, adequate assessment of potential confounders such as gestational age, birth weight, sepsis, or NEC would require complete data from each individual study to perform a meta-analysis of individual patient data, which is beyond our scope.

This information is now mentioned in the new version of the manuscript (lines 73-78).

Reviewer 2 Report

Comments and Suggestions for Authors

I read the paper entitled  “Platelet Counts and Risk of Severe Retinopathy of Prematurity: A Bayesian Model-Averaged Meta-analysis” very carefully and concluded that the paper is acceptable with minor revision for publication in your journal. Despite the fact that a few meta-analysis were already published in the last years (ref.3,4,7), the topic of the article is still interesting. The authors used a Bayesian model-averaged meta-analysis to confirm the association between platelet counts and ROP.

In the part Materials and Methods on page 2, Table 1 is present as Table S1, an error or what is S? the same is also in Results – Figure S1 on page 4. At the end on the page 2 there is a reference 24, which is also an error.

The references should be cited at the end of sentences.

Comments on the Quality of English Language

Minor editing of English language required

Author Response

In the part Materials and Methods on page 2, Table 1 is present as Table S1, an error or what is S? the same is also in Results – Figure S1 on page 4. At the end on the page 2 there is a reference 24, which is also an error.

The references should be cited at the end of sentences.

Answer:

Thank you for your comments and suggestions.

The “S” refers to supplementary table or Figure.

We have corrected the mistakes with the references.

Reviewer 3 Report

Comments and Suggestions for Authors

This paper is a meta-analysis of 21 studies concerning the association between low platelet counts and retinopathy of prematurity. The authors make the case for Bayesian meta-analysis, more appropriate for a small number of studies. There is also a detailed description of the statistical methods. 

I have particularly appreciated the use of RoBMA in order to assess the robustness of the results. It has actually modified consistently the final discussion, since only the association between severe ROP and thrombocytopenia showed a moderate evidence in the end.

I would like to ask what was the definition of lower platelet count (lines 146-147)

Also, at line 153 there is a typo, it should be mentioned that there are 5 studies presented in Figure C.

I believe that it is a sound research.

Author Response

  1. I would like to ask what was the definition of lower platelet count (lines 146-147).

Thank you very much for your comments and suggestions.

This refers to platelet counts as a continuous variable. This point has now been clarified in the new version of the manuscript.

  1. Also, at line 153 there is a typo, it should be mentioned that there are 5 studies presented in Figure C.

This has been corrected.

Round 2

Reviewer 1 Report

Comments and Suggestions for Authors

The authors have adequately responded to the comments.